# Transformer Architecture Search for Improving Out-of-Domain Generalization in Machine Translation

**Yiheng He** *  
*UC San Diego*

*yihenghe@berkeley.edu*

**Ruiyi Zhang** *  
*UC San Diego*

*ruz048@ucsd.edu*

**Sai Ashish Somayajula** *  
*UC San Diego*

*ssomayaj@ucsd.edu*

**Pengtao Xie**  
*UC San Diego*

*p1xie@ucsd.edu*

**Reviewed on OpenReview:** *https://openreview.net/forum?id=8M6cn6lfrJ*

## Abstract

Interest in automatically searching for Transformer neural architectures for machine translation (MT) has been increasing. Current methods show promising results in in-domain settings, where training and test data share the same distribution. However, in real-world MT applications, it is common that the test data has a different distribution than the training data. In these out-of-domain (OOD) situations, Transformer architectures optimized for the linguistic characteristics of the training sentences struggle to produce accurate translations for OOD sentences during testing. To tackle this issue, we propose a multi-level optimization based method to automatically search for neural architectures that possess robust OOD generalization capabilities. During the architecture search process, our method automatically synthesizes approximated OOD MT data, which is used to evaluate and improve the architectures' ability of generalizing to OOD scenarios. The generation of approximated OOD data and the search for optimal architectures are executed in an integrated, end-to-end manner. Evaluated across multiple datasets, our method demonstrates strong OOD generalization performance, surpassing state-of-the-art approaches. Our code is publicly available at `https://github.com/yihenghe/transformer_nas`.

## 1 Introduction

Machine Translation (MT) is a pivotal area within natural language processing (NLP), enabling the automatic translation of texts from one language to another. The emergence of Neural Machine Translation (NMT) (Bahdanau et al., 2014) has revolutionized the field by eliminating the need for labor-intensive feature engineering through an end-to-end training mechanism. Many deep neural networks, such as Recurrent Neural Networks (RNNs) Bahdanau et al. (2014) and Long Short-Term Memory (LSTM) Hochreiter & Schmidhuber (1997) networks, have propelled advancements in NMT. Notably, the Transformer model Vaswani et al. (2017), which leverages self-attention to capture long-range dependency between tokens, has significantly enhanced MT performance. Despite these advancements, current NMT models are manually designed,

---

*Equal contribution.

a process that demands significant time, resources, and expert knowledge, involving extensive trial-and-error for optimization. There may be superior design alternatives that go unexplored, hindered by constraints on time and resources, as well as potential human biases.

To tackle this issue, Neural Architecture Search (NAS) Liu et al. (2018); Chen et al. (2019) methods have been developed to autonomously identify the most effective Transformer architectures for machine translation. For example, Evolved Transformer So et al. (2019), which is an evolutionary algorithm based NAS method, achieves state-of-the-art MT performance. To mitigate the high computational costs incurred by the evolutionary algorithm in Evolved Transformer, Zhao et al. (2021) proposed a differentiable neural architecture search method which has the same space as Evolved Transformer. This approach significantly improves computation efficiency while achieving translation performance comparable to, if not better than, that of Evolved Transformer.

Despite their potential, existing NAS methods encounter a notable obstacle: limited out-of-domain generalization. Architectures identified through current NAS techniques perform well within the specific domains and datasets they are optimized for. However, their performance significantly decreases when dealing with texts from domains or styles that differ from those in the training set. The issue stems from the fact that NAS tends to overfit to the peculiarities of the training data, capturing idiosyncratic features that do not necessarily translate to broader linguistic or contextual generalizability. Consequently, when faced with novel or diverse linguistic data outside the narrow confines of their training environment, these specialized architectures often struggle to maintain the same level of accuracy and fluency, highlighting a critical gap between domain-specific optimization and universal applicability in machine translation tasks.

To address this problem, we propose a novel NAS framework to search for an optimal Transformer architecture for MT tasks that achieves superior OOD generalization performance. Our framework is based on multi-level optimization (MLO) Choe et al. (2022) which performs three learning stages end-to-end. In the first stage, we train the weight parameters of a Transformer model, keeping its learnable architecture tentatively fixed, by minimizing the loss on an MT training dataset. In the second stage, we generate an additional dataset that maximizes the distribution discrepancy with the real training data while preserving semantic consistency between paired source and target sentences. This step aims to create a close approximation to OOD data, addressing the limited availability of true OOD data during training. To achieve this, we introduce small, learnable perturbations to the embeddings of source sentences and optimize these perturbations. These adjusted source sentences, paired with their corresponding target sentences from the training dataset, form an auxiliary dataset that serves as synthetic OOD data. In the final stage, we assess the loss of the Transformer model (trained in the first stage) on the OOD data (generated in the second stage) and refine its architecture by minimizing this loss. The three stages are conducted jointly by solving the MLO problem. We conducted comprehensive experiments across both OOD and in-domain MT tasks. Our method significantly outperforms vanilla Transformer and prior NAS-based Transformer architectures. Extensive ablation studies further validate the importance of each component in our framework. Moreover, our method only incurs modest increase in computational costs (less than 10%), compared to baseline NAS methods.

To address this challenge, we propose a novel Neural Architecture Search (NAS) framework to identify optimal Transformer architectures for machine translation (MT) tasks with superior out-of-domain (OOD) generalization performance. Our approach leverages multi-level optimization (MLO) Choe et al. (2022), which operates in three integrated learning stages in an end-to-end manner.

The major contributions of this paper can be summarized as follows.

- We propose a three-level optimization based method that automatically searches for a Transformer architecture to improve the OOD generalization performance in MT.

- Our method significantly outperforms both the vanilla Transformer architecture and NAS baselines on OOD tasks. For instance, our approach enhanced the baseline method's performance by 29% in BLEU score when assessed using the Gnome (English-Igbo) dataset, and achieved a 38% improvement in BLEU score on the Ubuntu (English-Igbo) dataset.

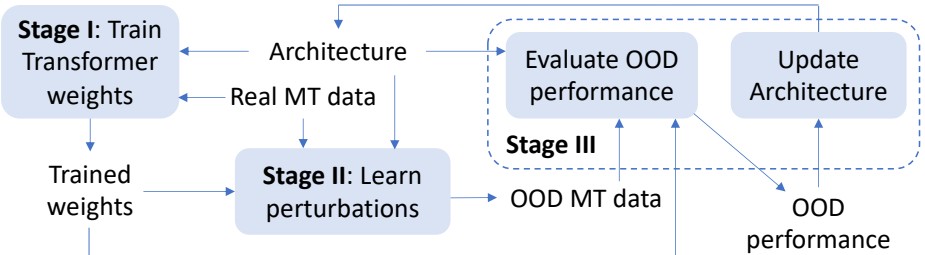

Figure 1: Overview of our method. It consists of three learning stages which are performed end-to-end. In the first stage, we train the Transformer's weight parameters with its architecture tentatively fixed. In the second stage, we generate a synthetic OOD MT dataset by adding learnable perturbations to real MT data. In the third stage, we evaluate the trained Transformer's performance on the OOD data and update its architecture by maximizing this performance.

## 2 Related Works

### 2.1 Neural Architecture Search (NAS)

The objective of NAS is to autonomously discover neural network architectures that have the potential to outperform those created by humans. In recent years, NAS has witnessed significant advancements. NAS methods are categorized into three primary types: reinforcement learning (RL)-based, evolutionary algorithms (EA)-based, and gradient-based. A detailed discussion of RL-based and EA-based NAS methods is provided in Appendix A. To address the issue of high computational cost of RL-based and EA-based NAS approaches, researchers have proposed differentiable search techniques Cai et al. (2019); Liu et al. (2018); Xie et al. (2019). These methods conceptualize each potential architecture as a configuration of numerous building blocks, with each block assigned a specific importance weight. The process of architecture search is then simplified to optimizing these weights, a task achievable through differentiable optimization techniques like gradient descent. Specifically, many gradient-based NAS methods use one-shot architecture search, in which architecture weights are learned by minimizing loss on a validation dataset while model weights are learned simultaneously by minimizing loss on a training dataset. This approach significantly enhances computational efficiency compared to RL-based methods. Recently, differentiable NAS techniques have been used to search for optimal Transformer architectures as well Zhao et al. (2021). Our method also belongs to the category of gradient-based NAS, but unlike previous approaches, it specifically focuses on searching for architectures that optimize out-of-domain generalization.

### 2.2 Out-of-Domain Generalizable Learning

Existing methods for improving the OOD generalizability of ML models can be roughly categorized into two paradigms. The first paradigm learns domain-invariant latent representations so that the discrepancy of different domains is mitigated in the latent space. Domain Invariant Component Analysis Muandet et al. (2013) is a kernel-based algorithm that learns a transformation to minimize the difference between marginal distributions of different domains. Multi-task Autoencoder (MTAE) Ghifary et al. (2015) jointly trains an encoder and several domain specific decoders to learn invariant representations with multi-task learning. MultiTCA Grubinger et al. (2017) learns domain-invariant representations by minimizing the discrepancy between domains while maximizing the variance within each domain. EISNet Wang et al. (2020a) learns to generalize across domains by solving a momentum metric learning task and a self-supervised auxiliary task.

The second paradigm of methods focuses on generating additional data and leveraging it to improve OOD generalizability. Domain Randomization Tobin et al. (2017) creates synthetic OOD images by randomly combining elements from different environment simulators. GUD Volpi et al. (2018) employs adversarial domain augmentation to iteratively generate fictitious worst-case target distributions. Jigen Carlucci et al. (2019) augments data by transforming it into jigsaw puzzles to simulate OOD scenarios. CrossGrad Shankar

et al. (2018) uses domain-guided perturbations to synthesize diverse training data. MADA Qiao et al. (2020) applies meta-learning to optimize meta parameters by minimizing the loss on augmented data from multiple domains and uses a Wasserstein autoencoder to relax semantic constraints commonly associated with domain augmentation. UMGUD Qiao & Peng (2021) introduces perturbations to input features and augments labels by injecting randomness derived from these perturbations. Unlike these methods, our approach employs multi-level optimization to integrate additional data simulation with OOD generalizable model training in an end-to-end manner. This holistic framework ensures that the simulated data is not only diverse but also highly effective in enhancing the model's robustness to distributional shifts.

### 2.3 Noisy Embeddings

It is worth noting that noisy embedding based data augmentation methods not only benefit OOD generalization tasks, as mentioned in the previous section, but enhance a model's in-domain capability as well. For instance, FreeLB Zhu et al. (2020) adds adversarial perturbations to word embeddings, enhancing the performance of Transformer-based models on natural language understanding and commonsense reasoning tasks. WaffleCLIP Roth et al. (2023) adds random words to image captions during the fine-tuning of vision-language models, yielding performance gains. NAYER Tran et al. (2023) introduces noisy layers to generate perturbed embeddings for image text captions, improving performance on data-free knowledge distillation tasks. BFTSS Somayajula et al. (2023) learns semantically consistent perturbations for embedding vectors, effectively enhancing performance on NLU tasks Wang (2018). NEFTune Jain et al. (2024) introduces random noise to the embedding vectors of the training data during the forward pass of fine-tuning, improving the outcome of instruction fine-tuning. Our method employs a novel strategy to generate noisy embeddings under a multi-level optimization framework, which enhances performance in both out-of-domain and in-domain scenarios.

### 2.4 Bi-level and Multi-level Optimization

A wide range of machine learning applications leverage bi-level optimization (BLO), which represents the simplest form of multi-level optimization, characterized by a two-tier hierarchy. Specifically, BLO consists of two nested optimization problems with a mutual dependency. These applications include, but are not limited to, neural architecture search Liu et al. (2018); Zhang et al. (2021), hyperparameter optimization Baydin et al. (2017); Feurer et al. (2015); Franceschi et al. (2017; 2018); Lorraine et al. (2020); Maclaurin et al. (2015), reinforcement learning Hong et al. (2020); Konda & Tsitsiklis (1999); Rajeswaran et al. (2020), data valuation Ren et al. (2020); Shu et al. (2019); Wang et al. (2020b), meta learning Finn et al. (2017); Rajeswaran et al. (2019), and label correction Zheng et al. (2019).

As BLO gains popularity, the focus has also broadened to multi-level optimization (MLO) involving more complex hierarchical structures Garg et al. (2022); He et al. (2021); Raghu et al. (2021); Somayajula et al. (2022); Such et al. (2020); Xie & Du (2022). MLO consists of more than two nested optimization problems with more complicated dependencies. Recent research in this area has shown a growing interest in constructing and optimizing multi-stage machine learning pipelines end-to-end using MLO. However, solving MLO problems poses computational challenges due to its complex structure. Sato et al. (2021) proposed a gradient-based solver. Choe et al. (2022) developed a software which enables users to compute hypergradients within MLO problems with multiple approximation methods easily and efficiently. To cope with high memory and computation costs associated with large-scale multi-level optimization, Choe et al. (2023) developed a distributed framework. In this paper, we propose a framework with three nested optimization problems, and solve this MLO problem with gradient-based algorithm.

## 3 Method

Let $W$ and $A$ denote the weight parameters and learnable architecture of a Transformer model used for machine translation (MT). This model takes a source sentence as input and generates a target sentence. Let $D^{tr} = \{(x_i, y_i)\}_{i=1}^N$ denote an MT training dataset where $x_i$ is an input source sentence and $y_i$ is

the corresponding target sentence. Our framework consists of three learning stages which are performed end-to-end.

## 3.1 Stage I

In the first stage, we train the weight parameters $W$ of the Transformer with its architecture $A$ tentatively fixed. Given a source sentence $x_i$ from $D^{tr}$, it is fed into the Transformer which generates a target sentence $f(E(x_i); W, A)$, where $E$ is an embedding module for sentences. A teacher-forcing based negative log likelihood (NLL) loss $l$ is used to measure the discrepancy between $f(E(x_i); W, A)$ and the ground-truth target sentence $y_i$. We learn $W$ by minimizing $l$ defined on each training example. This stage amounts to solving the following optimization problem:

$$W^*(A) = \text{argmin}_W \sum_{i=1}^{N} l(f(E(x_i); W, A), y_i) \tag{1}$$

where $W^*(A)$ denotes that the optimal solution $W^*$ depends on $A$ since $W^*$ depends on the loss function, which depends on $A$. Note that $A$ cannot be learned at this stage. Otherwise, the result will be an overfitted solution where $A$ perfectly fits the training data but performs poorly on the test data.

## 3.2 Stage II

In the second stage, we generate an additional MT dataset to learn an architecture with robust OOD generalization capabilities. Since real OOD data is unavailable, we create a close approximation of OOD data for use in architecture search in the next stage (see below). For each real training example $(x_i, y_i) \in D^{tr}$, we first map an input sentence $x_i$ to a sentence embedding with the embedding module $E$. We add a small learnable perturbation $\delta_i$ to $E(x_i)$ in a way that the perturbed embedding $E(x_i) + \delta_i$ satisfies the following two conditions. First, the translation $y_i$ for $E(x_i)$ can still be used as a translation for $E(x_i) + \delta_i$. Second, the perturbed source sentence embeddings $\widehat{S}(\{\delta_i\}_{i=1}^{N}) = \{E(x_i) + \delta_i\}_{i=1}^{N}$ should have a large domain discrepancy with the original source sentence embeddings $S = \{E(x_i)\}_{i=1}^{N}$. To satisfy the first condition, we use the trained MT model with weights $W^*(A)$ to generate a translation $f(E(x_i) + \delta_i; W^*(A), A)$ and learn $\delta_i$ to minimize the discrepancy (measured using the NLL $l$) between $f(E(x_i) + \delta_i; W^*(A), A)$ and $y_i$. To satisfy the second condition, we use the Maximum Mean Discrepancy (MMD) Gretton et al. (2012) $M$ to measure the domain difference between $\widehat{S}(\{\delta_i\}_{i=1}^{N})$ and $S$, and learn $\{\delta_i\}_{i=1}^{N}$ to maximize $M(\widehat{S}(\{\delta_i\}_{i=1}^{N}), S)$. Let $k(\cdot, \cdot)$ be a kernel function. Given two distributions $p$ and $q$, their MMD is defined as:

$$\mathbb{E}_{x \sim p, x' \sim p}[k(x, x')] + \mathbb{E}_{y \sim q, y' \sim q}[k(y, y')] - 2\mathbb{E}_{x \sim p, y \sim q}[k(x, y)]. \tag{2}$$

Given a set of sample $\{x_i\}_{i=1}^{m}$ drawn from $p$ and a set of sample $\{y_i\}_{i=1}^{n}$ drawn from $q$, the empirical MMD is calculated as:

$$\frac{1}{m(m-1)} \sum_{i=1}^{m} \sum_{j \neq i}^{m} k(x_i, x_j) + \frac{1}{n(n-1)} \sum_{i=1}^{n} \sum_{j \neq i}^{n} k(y_i, y_j) - \frac{2}{mn} \sum_{i=1}^{m} \sum_{j=1}^{n} k(x_i, y_j). \tag{3}$$

We choose to use MMD for measuring domain discrepancy because it is non-parametric, requiring no assumptions about the underlying distributions, and leverages the kernel trick to efficiently capture complex, high-dimensional data structures.

This stage amounts to solving the following optimization problem:

$$\{\delta_i^*(A)\}_{i=1}^{N} = \text{argmin}_{\{\delta_i\}_{i=1}^{N}} \sum_{i=1}^{N} l(f(E(x_i) + \delta_i; W^*(A), A), y_i) - \lambda M(\widehat{S}(\{\delta_i\}_{i=1}^{N}), S) \tag{4}$$

where $\lambda$ is a tradeoff parameter.

## 3.3 Stage III

In the third stage, we evaluate the MT model trained in the first stage on the generated dataset in stage II $\{(E(x_i) + \delta_i^*(A), y_i)\}_{i=1}^{N}$ and update the architecture $A$ by minimizing the evaluation loss. This stage

amounts to solving the following optimization problem:

$$\min_A \sum_{i=1}^N l(f(E(x_i) + \delta_i^*(A); W^*(A), A), y_i) \tag{5}$$

### 3.4 A Multi-level Optimization Framework

Putting these pieces together, we have the following three-level optimization problem:

$$\begin{aligned} \min_A \ & \sum_{i=1}^N l(f(E(x_i) + \delta_i^*(A); W^*(A), A), y_i) \\ s.t. \quad \{\delta_i^*(A)\}_{i=1}^N = & \operatorname{argmin}_{\{\delta_i\}_{i=1}^N} \sum_{i=1}^N l(f(E(x_i) + \delta_i; W^*(A), A), y_i) - \lambda M(\widehat{S}(\{\delta_i\}_{i=1}^N), S) \\ W^*(A) = & \operatorname{argmin}_W \ \sum_{i=1}^N l(f(E(x_i); W, A), y_i) \end{aligned} \tag{6}$$

The three stages are mutually dependent on each other. The output $W^*(A)$ of the first stage is the input of the loss function in the second stage. The outputs of the first two stages, including $W^*(A)$ and $\{\delta_i^*(A)\}_{i=1}^N$, are the inputs to the loss function in the third stage. The optimization variable $A$ in the third stage is used to define the loss functions in the first and second stage. By solving the three interdependent optimization problems together, we can perform the three learning stages end-to-end.

### 3.5 Search Space and Search Method

We set the the candidate operation set in our experiments following So et al. (2019); Zhao et al. (2021), which is detailed in Appendix B. Finally, we apply dropout Srivastava et al. (2014) to the output of an operation, then add this result to the original input to form a residual connection He et al. (2016). After this, layer normalization Ba et al. (2016) is applied to the combined output.

We adopt a differentiable search method Liu et al. (2018), where each candidate operation is associated with a selection variable representing how likely this operation is going to be selected into the final architecture. The architecture $A$ is represented by the collection of selection variables and architecture search amounts to learning these variables. After learning, operations associated with the highest-valued selection variables are preserved to construct the final architecture.

### 3.6 Optimization Algorithm

We develop a hypergradient based method to solve the problem in Eq.(10). First, we approximate $W^*(A)$ using one-step gradient descent update of $W$ w.r.t the loss function in the first stage:

$$W^*(A) \approx W' = W - \eta_w \nabla_W \sum_{i=1}^N l(f(E(x_i); W, A), y_i) \tag{7}$$

where $\eta_w$ is a learning rate. Then we plug the approximation $W^*(A) \approx W'$ into the loss function in the second stage, and approximate $\delta_i^*(A)$ using one-step gradient descent update of $\delta_i$ w.r.t the approximated loss function:

$$\delta_i^*(A) \approx \delta_i' = \delta_i - \eta_\delta \nabla_{\delta_i} \big( l(f(E(x_i) + \delta_i; W', A), y_i) - \lambda M(\widehat{S}(\{\delta_j\}_{j=1}^N), S) \big) \tag{8}$$

where $\eta_\delta$ is a learning rate. Finally, we plug the approximation $\delta_i^*(A) \approx \delta_i'$ into the loss function in the third stage and update $A$ by gradient descent:

$$A \leftarrow A - \eta_a \nabla_A \sum_{i=1}^N l(f(E(x_i) + \delta_i'; W', A), y_i) \tag{9}$$

where $\eta_a$ is a learning rate. After $A$ is updated, it is plugged into Eq.(7) and Eq.(8), yielding a new $W'$ and a new $\delta_i'$. These update steps iterate until convergence, yielding the final learned $A' \approx A^*$, as summarized in Algorithm 1. We provide additional details about our optimization algorithm in Appendix C, including the detailed computation of the gradient in Eq. (9) and a discussion on the convergence properties of the method.

---

**Algorithm 1** Optimization Algorithm

---

Initialize a model with weights $W$ and architecture $A$.
**while** *not converged* **do**
    1. Update weights $W$ with equation (7)
    2. Update perturbation $\delta$ with equation (8)
    3. Update architecture $A$ with equation (9)
**end while**
Derive the final architecture based on learned $A$.

---

## 4 Experiments

### 4.1 Datasets

We evaluate OOD generalization performance on low-resource languages, including English-Igbo (En-Ig), English-Hausa (En-Ha), and English-Irish (En-Ga) language pairs. Igbo, part of the Niger-Congo language family, incorporates diacritics, presenting challenges for MT tasks Orife (2018); Dossou & Emezue (2021). Hausa, a Chadic language within the Afroasiatic phylum, presents unique linguistic features. Irish, a Goidelic language of the Celtic family, employs an array of grammatical mutations and a VSO (verb-subject-object) word order, which pose challenges for machine translation tasks Dhonnchadha et al. (2003). Following Ahia et al. (2021), we obtain training data for En-Ig, En-Ha and En-Ga from the CCMatrix parallel corpus Schwenk et al. (2019), which offers the largest collection of high-quality, web-based bitexts for machine translation. The test data for En-Ig and En-Ha is from the Gnome and Ubuntu datasets, and the test data for En-Ga is from the Flores dataset, all considered OOD for CCMatrix. We include detailed description of these datasets in Appendix D.

Furthermore, we conduct experiments on high-resource languages, following So et al. (2019) and Zhao et al. (2021), using the following training datasets: 1) WMT18 English-German (En-De) without ParaCrawl, consisting of 4.5 million sentence pairs; 2) WMT14 English-French (En-Fr), comprising 36 million sentence pairs; and 3) WMT18 English-Czech (En-Cs) without ParaCrawl, with 15.8 million sentence pairs. The test data for these language pairs is from the WMT-Chat and WMT-Biomedical datasets, both considered OOD for WMT14 and WMT18. Detailed descriptions can be found in Appendix D.

### 4.2 Baselines

Our method is evaluated against the vanilla Transformer architecture Vaswani et al. (2017). We also compare our method with differentiable architecture search baselines including DARTS Liu et al. (2018) and PDARTS Chen et al. (2019), that balance performance and computational efficiency. We did not compare with evolutionary algorithm based methods So et al. (2019) since they are computationally very expensive. We include detailed description of baseline methods in Appendix E.

Our method represents a general framework that can be integrated with various differentiable NAS methods. For example, the search space and search strategy in our method can be set to those in either DARTS or PDARTS. We use Ours-darts and Ours-pdarts to denote that our method is integrated with DARTS and PDARTS respectively.

### 4.3 Experimental Settings

**Search configuration.** We follow the settings in Zhao et al. (2021) for architecture search. Both the vanilla DARTS and Ours-darts utilize two identical encoder and two identical decoder layers during the search phase. Each encoder layer consists of 'Self Attention' → 'Search Node' → 'Search Node', whereas each decoder layer consists of 'Self Attention' → 'Cross Attention' → 'Search Node' → 'Search Node'. Only the 'Search Node' is searched, while the architectures of 'Self Attention' and 'Cross Attention' are fixed. Zhao et al. (2021) empirically shows that architecture search with this configuration yields better results. The layers with searched architecture are stacked to construct the final model with six encoder layers and

| Methods | En-Ig (Gnome) | En-Ig (Ubuntu) | En-Ha (Gnome) | En-Ha (Ubuntu) | En-Ga (Flores) | En-De (Chat) | En-De (Biomed) | En-Fr (Chat) | En-Cs (Biomed) |
|---|---|---|---|---|---|---|---|---|---|
| Transformer | $2.53_{(0.07)}$ | $2.04_{(0.07)}$ | $0.99_{(0.12)}$ | $0.52_{(0.12)}$ | $30.79_{(0.20)}$ | $28.71_{(0.48)}$ | $54.08_{(0.11)}$ | $26.90_{(0.58)}$ | $19.84_{(0.26)}$ |
| DARTS | $1.22_{(0.26)}$ | $1.05_{(0.39)}$ | $0.75_{(0.04)}$ | $0.38_{(0.02)}$ | $26.63_{(0.08)}$ | $29.03_{(0.25)}$ | $55.39_{(0.94)}$ | $18.30_{(0.26)}$ | $22.62_{(0.45)}$ |
| Ours-darts | $\mathbf{2.97_{(0.01)}}$ | $\mathbf{2.39_{(0.07)}}$ | $\mathbf{1.41_{(0.07)}}$ | $\mathbf{0.83_{(0.08)}}$ | $\mathbf{33.19_{(0.20)}}$ | $\mathbf{30.21_{(0.57)}}$ | $\mathbf{58.71_{(0.46)}}$ | $\mathbf{28.06_{(0.20)}}$ | $\mathbf{25.08_{(0.38)}}$ |
| PDARTS | $1.28_{(0.25)}$ | $1.43_{(0.51)}$ | $0.96_{(0.07)}$ | $0.49_{(0.14)}$ | $29.56_{(0.25)}$ | $29.60_{(1.27)}$ | $58.24_{(0.64)}$ | $27.41_{(0.22)}$ | $23.53_{(0.27)}$ |
| Ours-pdarts | $\mathbf{3.27_{(0.20)}}$ | $\mathbf{2.81_{(0.21)}}$ | $\mathbf{1.58_{(0.08)}}$ | $\mathbf{1.01_{(0.07)}}$ | $\mathbf{33.33_{(0.24)}}$ | $\mathbf{31.29_{(0.80)}}$ | $\mathbf{59.73_{(0.94)}}$ | $\mathbf{28.09_{(0.42)}}$ | $\mathbf{24.91_{(0.28)}}$ |

Table 1: BLEU scores (mean and standard deviation across three runs) in nine out-of-domain (OOD) generalization experiments, where each column corresponds to an experimental setting: (1-2) The training dataset was En-Ig (CCMatrix) and the test datasets included En-Ig (Gnome) and En-Ig (Ubuntu); (3-4) The training dataset was En-Ha (CCMatrix) and the test datasets included En-Ha (Gnome) and En-Ha (Ubuntu); (5) The training dataset was En-Ga (CCMatrix) and the test dataset was En-Ga (Flores); (6-7) The training dataset was En-De (CCMatrix) and the test datasets included En-De (WMT-Chat) and En-De (WMT-Biomedical); (8) The training dataset was En-Fr (CCMatrix) and the test dataset was En-Fr (WMT-Chat); and (9) The training dataset was En-Cs (CCMatrix) and the test dataset was En-Cs (WMT-Biomedical). Double-sided t-tests were conducted between our methods and baselines, with p-values less than 0.05, indicating statistically significant improvements achieved by our methods over baselines.

six decoder layers. PDARTS and Ours-pdarts adopt a progressive learning approach, where the number of encoder and decoder layers increases from 2 to 4 to 6 through the search process. Simultaneously, the size of the operation set in the encoder layer is reduced from 15 to 10 to 5, and in the decoder layer from 16 to 11 to 6.

**Hyperparameter settings.** Following Vaswani et al. (2017), we utilize 6 encoder and decoder layers, a hidden size of 512, a filter size of 2048, and 8 attention heads for vanilla Transformers, DARTS, PDARTS, Ours-darts and Ours-pdarts. For Ours-darts and Ours-pdarts, the radial basis function (RBF) kernel is used to compute maximum mean discrepancy (MMD) used in Stage II. The tradeoff parameter $\lambda$ is set to 1.5.

For the optimization of $W$ in our methods and baselines, the same hyperparameter settings as Vaswani et al. (2017) are used, including the learning rate and its scheduler, with warm-up steps set to 4000. The Adam Kingma & Ba (2014) optimizer with $\beta_1 = 0.9$, $\beta_2 = 0.98$, and $\epsilon = 10^{-9}$ is used. We increase the learning rate linearly for the first warmup training steps, and decrease it thereafter proportionally to the inverse square root of the step number. For the optimization of architecture weights $A$, both our methods and the baselines use the same hyperparameters following Liu et al. (2018), employing a constant learning rate of $3 \times 10^{-4}$ and a weight decay of $10^{-3}$, with the Adam optimizer ($\beta_1 = 0.9$, $\beta_2 = 0.98$, and $\epsilon = 10^{-9}$) being used. The perturbation $\delta$, in our framework, is optimized using an Adam optimizer with $\beta_1 = 0.9$, $\beta_2 = 0.98$, and $\epsilon = 10^{-9}$ and a constant learning rate of $10^{-3}$. We set the dropout rate to 0.1 and employ label smoothing with a value of 0.1 during architecture search. We set the batch size to 4096 for all experiments.

The maximum sentence length is set to 256 for all experiments. Tokenization is performed using Moses[1], which is a rule-based tokenizer. BLEU Papineni et al. (2002) is used as the evaluation metric. We employ beam search during inference with a beam size of 4 and a length penalty $\alpha = 0.6$. All the experiments were conducted on Nvidia A100 GPU.

## 4.4 Results on Out-of-Domain Generalization

We evaluate the OOD generalization performance of our method in nine experiments. In experiment 1 and 2, the training dataset was En-Ig (CCMatrix) and the test datasets included En-Ig (Gnome) and En-Ig (Ubuntu). In experiment 3 and 4, the training dataset was En-Ha (CCMatrix) and the test datasets included En-Ha (Gnome) and En-Ha (Ubuntu). In experiment 5, the training dataset was En-Ga (CCMatrix) and

---

[1]https://github.com/moses-smt/mosesdecoder.git

| Methods | En-Fr (WMT) | En-Cs (WMT) |
|---|---|---|
| Transformer | 38.14 | 24.22 |
| DARTS | 38.42 | 25.00 |
| Ours-darts | **38.85** | **25.24** |
| PDARTS | 39.39 | 25.33 |
| **Ours-pdarts** | **39.73** | **25.89** |

Table 2: BLEU scores in the third OOD generalization experiment, where the dataset for architecture search was En-De (WMT) and the datasets for retraining and testing included En-Fr (WMT) and En-Cs (WMT).

| Methods | En-Ig (CCmatrix) | En-Ha (CCmatrix) | En-De (WMT) |
|---|---|---|---|
| Transformer | $52.04 \pm 0.59$ | $44.73 \pm 1.05$ | 27.27 |
| DARTS | $41.95 \pm 1.04$ | $37.22 \pm 2.33$ | 27.55 |
| Ours-darts | $\mathbf{55.50} \pm 0.18$ | $\mathbf{47.28} \pm 0.94$ | **27.84** |
| PDARTS | $45.38 \pm 1.01$ | $40.34 \pm 0.62$ | 28.11 |
| **Ours-pdarts** | $\mathbf{59.23} \pm 2.09$ | $\mathbf{57.41} \pm 1.90$ | **28.24** |

Table 3: BLEU scores of in-domain generalization experiments, where training and test data are from the same dataset. Each of the three datasets, including En-Ig (CCmatrix), En-Ha (CCmatrix), and En-De (WMT), is divided into training, validation, and test splits. Models are trained on the training split, evaluated on the test split (results shown in this table), and the validation split is used for tuning hyperparameters.

the test dataset was En-Ga (Flores). In experiments 6 and 7, the training dataset was En-De (CCMatrix) and the test datasets included En-De (WMT-Chat) and En-De (WMT-Biomedical). In experiment 8, the training dataset was En-Fr (CCMatrix) and the test dataset was En-Fr (WMT-Chat). In experiment 9, the training dataset was En-Cs (CCMatrix) and the test dataset was En-Cs (WMT-Biomedical). Additionally, we evaluate the OOD generalization performance of our method across language pairs. Specifically, the dataset for architecture search was En-De (WMT) and the datasets for retraining and testing included En-Fr (WMT) and En-Cs (WMT).

Table 1 and 2 show the results. As can be seen, Ours-darts and Ours-pdarts demonstrate superior performance compared to DARTS and PDARTS, respectively. In particular, Ours-pdarts achieves the best performance on 8 out of 9 datasets across all methods. This is because our method automatically generates approximated OOD data (in stage II) and optimizes the architecture by minimizing the MT loss on this generated data (in stage III). By doing so, our method aims to enhance the architecture's ability to generalize to real OOD data. Such a mechanism is lacking in DARTS and PDARTS. Furthermore, Ours-darts and Ours-pdarts surpass the performance of the vanilla Transformer architecture, which further demonstrates the superior OOD generalization capability of our methods, due to its mechanism of generating additional data to improve the OOD generalization performance of architectures. In contrast, the manual designed Transformer architecture does not take OOD generalization into account.

## 4.5 Results on In-Domain Generalization

In addition to the OOD generalization performance, we also evaluated the in-domain generalization performance of searched architectures, where the training and test data are disjoint splits of the same dataset. Datasets used for this evaluation include WMT18 English-German (En-De), WMT14 English-French (En-Fr), WMT18 English-Czech (En-Cs), CCMatrix English-Igbo (En-Ig) and CCMatrix English-Hausa (En-Ha). Table 3 shows the results. As can be seen, Ours-darts outperforms DARTS; Ours-pdarts outperforms PDARTS; and both of our methods outperform vanilla Transformer. This demonstrates the superior per-

formance of our methods in scenarios where the domain of language pairs in test data are the same as those in the training data. This superiority highlights the robustness of our method to both in-domain and out-of-domain machine translation scenarios.

## 4.6 Ablation Studies

To better evaluate the effectiveness of individual components in our framework, we perform several ablation studies.

**Sensitivity to the tradeoff parameter $\lambda$.** We investigate how the trade-off parameter $\lambda$ in our framework affects downstream performance. $\lambda$ is varied within the set $\{0.5, 1.0, 1.5, 2.0\}$. The WMT18 English-German dataset was used as the training data and WMT14 English-French was used as test data. The study was conducted using Ours-darts. Figure 2 shows how the BLEU score on the WMT14 English-French test set varies as $\lambda$ increases. As can be seen, a $\lambda$ value in the middle ground yields the best performance. A very small $\lambda$ such as 0.5 reduces the contribution from the MMD, making the domain difference between generated data and real training data small. As a result, the generated data cannot effectively improve the OOD generalization performance of the architecture. Conversely, a higher $\lambda$ increases the contribution from the MMD, encouraging a larger domain difference but with the risk of deviating too much such that the translation of $E(x_i) + \delta_i$ no longer corresponds to $y_i$. Thus, achieving a balance is essential by choosing an optimal $\lambda$. From the results, we observe that $\lambda = 1.5$ is optimal.

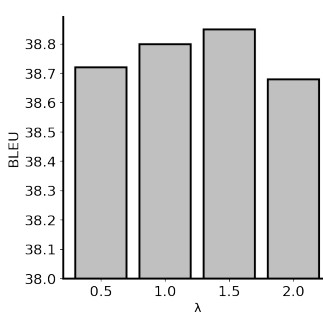

Figure 2: BLEU score of Ours-darts on WMT (En-Fr) with varying $\lambda$.

| Methods | En-Fr | En-Cs |
|---|---|---|
| DARTS | 38.42 | 25.00 |
| No-MMD-darts | 37.70 | 24.30 |
| $L_2$-darts | 38.64 | 25.04 |
| No-Stage-II-darts | 38.66 | 25.09 |
| Ours-darts | **38.85** | **25.24** |
| PDARTS | 39.39 | 25.33 |
| No-MMD-pdarts | 37.85 | 25.09 |
| $L_2$-pdarts | 39.49 | 25.44 |
| No-Stage-II-pdarts | 39.53 | 25.49 |
| Ours-pdarts | **39.73** | **25.89** |

Table 4: BLEU scores on the test set of WMT English-French (En-Fr) and English-Czech (En-Cs) datasets, under various ablation study settings: 1) removing the MMD loss term (No-MMD), 2) replacing MMD loss with L2 loss (L2), and 3) removing the second stage from our framework (No-Stage-II).

**Impact of the MMD loss term.** We study the impact of the MMD loss term in our framework and its effect on downstream performance. This experiment is carried out by 1) omitting the MMD loss term (denoted as *No-MMD*) and 2) replacing the MMD loss term with an $L_2$ regularization on $\delta$ (denoted as $L_2$). The training data was WMT14 English-German. The test data was WMT14 English-French and WMT18 English-Czech. Table 4 shows the results.

We observe that No-MMD-darts and No-MMD-pdarts underperform compared to Ours-darts and Ours-pdarts, respectively, due to the absence of the MMD loss term. This absence leads to a degenerate solution of $\delta = 0$, resulting in the generated additional data identical to the real training data $D^{tr}$ and increasing the risk of overfitting as both model weights and architecture parameters are optimized on $D^{tr}$. This is evident from the inferior performance of No-MMD-darts and No-MMD-pdarts compared to DARTS and PDARTS, which learn model weights and architecture parameters on disjoint splits of the training dataset to prevent overfitting. The results underscore the critical role of the MMD loss in creating a significant domain gap

| Methods | DARTS-ood | Ours-darts |
|---|---|---|
| **En-Ig (Ubuntu)** | $2.20 \pm 0.05$ | $\mathbf{2.39} \pm 0.07$ |

Table 5: BLEU score comparison between DARTS-ood and Ours-darts, which use real OOD validation data and generated OOD data to optimize architectures respectively. BLEU scores are measured on the Ubuntu (En-Ig) dataset.

between the generated MT data and $D^{tr}$, which is crucial for improving the OOD generalization performance of searched architectures subsequently.

Furthermore, we observe that Ours-darts and Ours-pdarts outperform $L_2$-darts and $L_2$-pdarts, respectively, on both language pairs. This indicates that MMD outperforms the L2 distance in quantifying and amplifying the differences between domains. The calculation of MMD involves all data examples from both datasets, thereby more accurately reflecting dataset differences at a distributional level. It achieves this by calculating the kernel-based dissimilarity for each pair of data examples, both within and between datasets, then combining these dissimilarity measures to form a comprehensive MMD score. In contrast, the L2 distance measures are limited to comparing pairs comprising a real sentence and its perturbed counterpart, focusing solely on individual data example disparities rather than assessing the dataset level difference. On the other hand, $L_2$-darts and $L_2$-pdarts outperform vanilla DARTS and PDARTS. Although L2 distance is not as effective as MMD, it can still enlarge the domain difference between generated data and real data, thereby improving architectures' OOD generalization performance.

**Impact of removing Stage II.** In this study, we explore the significance of Stage II within our framework by removing it (denoted as *No-Stage-II*). Rather than acquiring the perturbations through learning, we randomly generate Gaussian noise and employ this as the perturbations. These are then added to the embeddings of real MT sentences to create the synthetic OOD data for neural architecture search. After removing Stage II, the original three-level optimization framework is reduced to a bi-level formulation:

$$
\begin{aligned}
&\min_A \; \sum_{i=1}^{N} l(f(E(x_i) + \widehat{\delta}_i(A); W^*(A), A), y_i) \\
&s.t. \; W^*(A) = \operatorname{argmin}_W \; \sum_{i=1}^{N} l(f(E(x_i); W, A), y_i)
\end{aligned}
\tag{10}
$$

where $\widehat{\delta}_i$ is drawn from a Gaussian distribution with a mean of $2.7689 \times 10^{-5}$ and a standard deviation of 0.0026). These parameters were derived from the statistics of the learned $\delta^*$ in Ours-darts.

The results, presented in Table 4, reveal that both No-Stage-II-darts and No-Stage-II-pdarts exhibit decreased performance compared to their counterparts Ours-darts and Ours-pdarts. This outcome strongly underscores the importance of Stage II, which learns perturbations instead of setting them randomly. In Ours-darts and Ours-pdarts, the perturbations are learned to create the 'optimal' synthetic OOD data that is best suitable for evaluating and improving the OOD generalization performance of architectures. In contrast, the generated approximated OOD data in No-Stage-II-darts and No-Stage-II-pdarts is randomly generated, which is sub-optimal. Notably, No-Stage-II-darts and No-Stage-II-pdarts still outperform vanilla DARTS and PDARTS, suggesting that even randomly generated perturbations to create additional data can improve the OOD generalization performance of architectures, compared to not using OOD data at all.

**Impact of generating out-of-domain validation data.** We investigate the effect of generating an additional dataset to approximate OOD data for architecture search within our framework, rather than relying on pre-existing OOD datasets. For this ablation study, we conducted experiments on the English-Igbo language pair. We used CCMatrix as the training set and Gnome as the OOD validation set to perform architecture search using DARTS. Ubuntu was used as the test set. The results, presented in Table 5, show that our method outperforms this ablation setting, highlighting the importance of generating synthetic OOD dataset within our framework. Compared with a fixed OOD dataset, the generated dataset can be more diverse since it is explicitly optimized to be OOD. This increased diversity can lead to more robust OOD generalization performance.

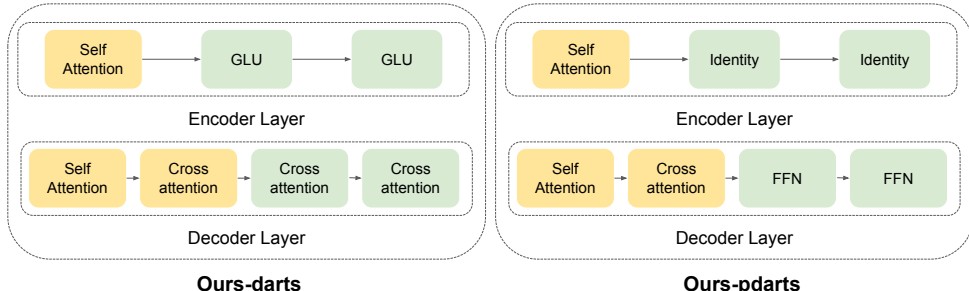

Figure 3: Architectures searched by Ours-darts and Ours-pdarts on the WMT (En-De) dataset. Each architecture consists of encoder layers and decoder layers. Multiple identical layers with searched architecture are stacked to construct the final model. Green boxes represent search nodes that are optimized during the search phase, while yellow boxes indicate predefined layers which are fixed during the search process.

| Methods | En-De | En-Fr | En-Cs |
|---|---|---|---|
| Ours-darts | 27.84 | 38.85 | 25.24 |
| Ours-darts$_L$ | **28.79** | **41.64** | **26.44** |
| Ours-pdarts | 28.24 | 39.73 | 25.89 |
| Ours-pdarts$_L$ | **29.29** | **42.97** | **27.26** |

Table 6: BLEU score comparison between Ours-darts, Ours-pdarts and their larger architecture versions Ours-darts$_L$ and Ours-pdarts$_L$, across En-De, En-Fr, and En-Cs language pairs, demonstrating the benefits of model scaling.

**Impact of model size.** To investigate the impact of final model size on machine translation performance, we conducted search to develop larger architectures, denoted as Ours-darts$_L$ and Ours-pdarts$_L$. Following the 'Transformer-big' architecture design in Vaswani et al. (2017), our large model architecture consists of 6 encoder and decoder layers, a hidden size of 1024, a filter size of 4096, and 16 attention heads. Evaluations were conducted across three language pairs: English-German, English-French, and English-Czech, as detailed in Table 6. The results demonstrate that increasing the final model size significantly improves translation performance on these language pairs, underscoring the importance of model scaling in achieving higher translation accuracy.

## 5 Discussion

**In-domain generalization performance.** The superior in-domain generalization performance of our methods can be attributed to several interrelated factors. Firstly, synthesizing approximated OOD data during the architecture search phase exposes the model to a wider array of linguistic features, enhancing its ability to generalize across diverse scenarios. This exposure likely leads to the selection of more robust neural architectures that are inherently better at handling not only OOD but also in-domain data. Additionally, training with OOD data may serve as a form of regularization, preventing overfitting and promoting a more generalized understanding of the language, which in turn facilitates in-domain generalization.

**Model size and computational costs** We compare the total number of parameters for our method and all baselines in Table 7. Ours-darts has fewer parameters than DARTS and Transformer. Ours-pdarts has fewer parameters than PDARTS. This indicates that our method significantly improves the OOD MT performance without increasing model size. The computational costs of conducting architecture search with Ours-darts, DARTS, Ours-pdarts, and PDARTS are presented in Table 8, in terms of GPU days. Our methods, including Ours-darts and Ours-pdarts, do not incur significantly higher costs (less than 10%) compared to baseline methods. This marginal increase in computational expense is notably outweighed by the significant performance enhancements observed across various OOD machine translation tasks.

| Methods | Transformer | DARTS | Ours-darts | PDARTS | Ours-pdarts |
|---|---|---|---|---|---|
| **Parameters** | 60.8M | 62.8M | 54.5M | 73.4M | 73.4M |

Table 7: Number of model parameters.

| Methods | DARTS | Ours-darts | PDARTS | Ours-pdarts |
|---|---|---|---|---|
| **Cost** | 3.13 | 3.40 | 9.26 | 9.59 |

Table 8: Comparison of search costs in GPU days for baseline methods DARTS, PDARTS, and our methods Ours-darts and Ours-pdarts.

**Analysis of searched architectures** We analyze the architectures searched by our methods, shown in Figure 3, with green boxes indicating the search nodes optimized during the search phase, and yellow boxes representing nodes with predefined architectures. The architecture learned by Ours-darts utilizes self-attention for input processing and GLUs for selective feature enhancement in the encoder, while its decoder refines the output iteratively with multiple cross-attention layers. In contrast, the architecture searched by Ours-pdarts, which yields the best results, features a streamlined encoder that employs self-attention for feature extraction. The extracted features are further maintained through identity layers. The decoder is more complex, incorporating both self-attention for refining its own output and cross-attention for integrating encoder information, further enriched by two FFNs for improved feature processing. Interestingly, the searched architectures opt to exclude convolution operations, implying that the self-attention and cross-attention mechanisms adequately capture the relevant features for the MT task. This effectiveness is likely augmented by operations such as GLUs and FFNs, which improve feature representation without the need for convolutions. This observation is in line with the architecture choices of popular Transformer models that also omit convolution operations Devlin et al. (2018); Liu et al. (2019); Lewis et al. (2019); He et al. (2020); Yang et al. (2019); Brown et al. (2020). Further comparisons between the architectures searched by our method and those found by baseline methods are provided in Appendix F.

# 6 Conclusions and Future Works

In this paper, we propose a novel multi-level optimization framework for searching OOD-generalizable Transformer architectures for MT tasks. Our method automatically generates additional data from the available training data that approximates OOD data, which is subsequently used to enhance the OOD generalization performance of the searched architectures. Our framework consists of three optimization stages performed end-to-end: 1) Training Transformer model weights on original real MT training data, 2) Generating additional synthetic OOD MT data that diverges in domain from the real training dataset, 3) Searching for the Transformer architecture that minimizes loss on this generated approximated OOD MT data. We demonstrate the effectiveness of our method across a variety of OOD machine translation tasks. Furthermore, our searched architectures also achieve high performance on various in-domain MT tasks. Moreover, through various ablation studies, we further highlight the importance of learning adversarial perturbations through our formulation to generate additional synthetic OOD data for OOD generalizable Transformer architecture search.

In this work, our focus has been on applying our method to MT tasks, with plans to extend it to a variety of NLP tasks such as text classification, named entity recognition, and text summarization. Additionally, we are exploring its applicability to other modalities, including image and audio data. This broadened scope aims to enhance our method's versatility and robustness in addressing OOD generalizable architecture search across diverse tasks and modalities.

# 7 Acknowledgement

This research was supported by NSF IIS2405974 and NSF IIS2339216.

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

## A  RL-based and EA-based Neural Architecture Search Methods

Early NAS methodologies Zoph & Le (2016); Pham et al. (2018); Zoph et al. (2018) are based on reinforcement learning (RL). They train a policy network to generate optimal architectures by maximizing validation accuracy, which serves as the reward. These strategies are straightforward in principle and offer the flexibility to explore various search spaces. Nonetheless, they require substantial computational resources. Specifically, to evaluate the performance of a proposed architecture, it necessitates training on a dataset, a process that is notably resource-intensive and time-consuming.

Another category of NAS methods Liu et al. (2017); Real et al. (2019) employ evolutionary algorithms (EA). In these methods, architectures are treated as individuals within a population, each rated by a fitness score that reflects its performance. Architectures that score higher are more likely to produce offspring (i.e., new architectures), which then supplant those with lower fitness scores. Similar to RL-based methods, EA-based methods are also marked by high computational demands, as they require the training of each architecture to evaluate its fitness score. Recently, EA-based NAS methods have been applied to search for optimal architectures of Transformers So et al. (2019), and achieved state-of-the-art performance on multiple machine translation tasks.

## B  Candidate Operation Set

Following So et al. (2019); Zhao et al. (2021), the candidate operation set in our experiments includes:

- Feed-Forward Neural Network (FFN)

- Self Attention: head $= 8$

- Identity: no transformation applied to the input

- Standard Convolution: $w \times 1$ with $w \in \{1, 3\}$

- Depth-wise Separable Convolution: $w \times 1$ with $w \in \{3, 5, 7, 9, 11\}$

- Dynamic Convolution: $w \times 1$ with $w \in \{3, 7, 11, 15\}$

- Gated Linear Unit (GLU)

- Cross Attention: head $= 8$, only available to decoder

## C  Optimization Algorithm

We first provide a detailed computation of the gradient used in our optimization algorithm. In Eq.(9), the gradient is calculated as follows:

$$\nabla_A l(f(E(x_i) + \delta_i'; W', A), y_i)$$
$$= \nabla_A \delta_i' \left( \frac{\partial l(f(E(x_i) + \delta_i'; W', A), y_i)}{\partial \delta_i'} \right) + \nabla_A W' \left( \frac{\partial l(f(E(x_i) + \delta_i'; W', A), y_i)}{\partial W'} \right) + \frac{\partial l(f(E(x_i) + \delta_i'; W', A), y_i)}{\partial A} \tag{11}$$

where $\frac{\partial \cdot}{\partial \cdot}$ denotes partial derivative. $\nabla_A \delta_i'$ can be calculated as:

$$\nabla_A \delta_i' = -\eta_\delta \nabla_{A, \delta_i}^2 l(f(E(x_i) + \delta_i; W', A), y_i) \tag{12}$$

$\nabla_A W'$ can be calculated as:

$$\nabla_A W' = -\eta_w \nabla_{A, W}^2 \sum_{i=1}^{N} l(f(E(x_i); W, A), y_i). \tag{13}$$

|       | CCMatrix | Gnome | Ubuntu |
|-------|----------|-------|--------|
| En-Ig | 5.9M     | 3173  | 608    |
| En-Ha | 80.4K    | 998   | 219    |

Table 9: Number of sentences in the CCMatrix, Gnome, and Ubuntu datasets for the English-Igbo (En-Ig) and English-Hausa (En-Ha) language pairs.

We now discuss key properties of our proposed optimization algorithm. Previous research has extensively analyzed the convergence properties of optimization algorithms for bi-level optimization (BLO) problems. For instance, Franceschi et al. (2018) established sufficient conditions under which solutions of approximate BLO problems converge to those of exact problems. Similarly, Shaban et al. (2018) provided sufficient conditions for the convergence of approximate gradients obtained via truncated back-propagation during iterative optimization. Further, convergence analyses for implicit gradient-based methods have been explored in works such as Grazzi et al. (2020) and Zhang et al. (2021). Experimental results in this paper demonstrate that our proposed algorithm consistently converges and achieves superior performance compared to baseline methods. However, our algorithm diverges from traditional BLO frameworks due to its multi-level optimization (MLO) structure, introducing unique challenges and opportunities for future exploration of its convergence properties.

Although approximated optimization algorithms for MLO problems, including our method, are theoretically and empirically likely to converge to stationary points, it remains unclear whether they achieve global optima rather than sub-optimal solutions. Nevertheless, prior research has demonstrated that sub-optimal solutions can still yield superior generalization performance (Dinh et al., 2017; Li et al., 2017), a finding corroborated by our experimental results. Furthermore, existing studies suggest that MLO algorithms enhance generalizability and mitigate overfitting, attributed to their multi-stage optimization process and the use of distinct data splits (Somayajula et al., 2022; Bao et al., 2021). Specifically, by optimizing different sets of parameters on distinct data subsets, MLO-based methods effectively mitigate overfitting to a single dataset (Zhang et al., 2024).

## D    Datasets

The Gnome dataset comprises 187 language pairs derived from the translation of Gnome documentation [2], while the Ubuntu dataset includes 42 language pairs generated from the translation of Ubuntu OS localization files [3]. Each sentence in the Gnome and Ubuntu datasets has an average length of 6-9 tokens. Table 9 summarizes the dataset statistics. WMT-Chat dataset includes data for translating conversational text, in particular customer support chats [4]. WMT-Biomedical dataset includes data for the translation of documents from the biomedical domain [5]. Flores dataset is a multilingual machine translation benchmark, which consists of translations primarily from English into around 200 language varieties [6].

Here, we further discuss the discrepancies among the CCMatrix, Gnome, and Ubuntu datasets. Gnome contains technical and UI-related text specific to the Gnome environment, while Ubuntu contains system messages and interface text in Ubuntu OS, representing a software domain. The text in this domain includes specific terms ("permissions", "kernel", "GUI", "repository", "configuration file"), technical verbs ("execute," "compile," "deploy.") and abbreviations ("sudo," "bash," "root"). Sentences are typically task-oriented or informational, dealing with instructions, warnings, and descriptions of software functionality.

In contrast, CCMatrix is derived from multilingual web pages which cover a wide range of topics, representing a general domain. For text in the general domain, it includes topics such as politics, sports, entertainment,

---

[2] https://www.gnome.org

[3] https://ubuntu.com

[4] https://wmt-chat-task.github.io/

[5] https://www2.statmt.org/wmt24/biomedical-translation-task.html

[6] https://github.com/openlanguagedata/flores

and daily conversations. The word usage is diverse, including those related to politics (`"democracy"`), culture (`"art gallery"`), and emotions (`"love"`), which would be uncommon in software datasets.

To sum up, the difference between CCMatrix and Gnome/Ubuntu datasets differs from multiple aspects including vocabulary and style, making them ideal datasets to investigate out-of-domain generalization performance of machine translation models.

## E  Baseline Methods

In the DARTS framework Liu et al. (2018), each search layer comprises multiple search nodes and the objective is to determine the most suitable operation for each node from a predefined candidate set, denoted as $O$. This set includes operations such as FFNs, self-attention mechanisms, and so on (see Appendix B). Each operation is applied to either the input of the layer or the outputs from intermediate nodes, generating new outputs for subsequent processing. DARTS represents the output of each search node as a weighted sum of outputs of all operations in $O$. The weights for each operation in this sum, pertinent to a search node, are derived from the softmax of learnable parameters, referred to as architecture weights $A$. They act as selection variables, representing the likelihood of this operation being selected into the final architecture. Specifically, for the output of a search node $f(\cdot)$ and input $x$, the operation is defined as,

$$f(x) = \sum_{o \in O} \frac{\exp(\alpha_o)}{\sum_{o' \in O} \exp(\alpha_{o'})} o(x) \tag{14}$$

where $O$ is the candidate operation set, $o \in O$ is an operation within this candidate set and $\{\alpha_o\}$ are the architecture weights for the search node. DARTS utilizes a bi-level optimization framework to learn architecture weights $A$ and model parameters $W$ on two disjoint splits $D_1$ and $D_2$ of the training dataset. $W$ is optimized by minimizing a loss $L$ on data split $D_1$ in the lower level, and $A$ is learned on data split $D_2$ in the upper level:

$$\min_A L(W^*(A), A, D_2) \tag{15}$$

$$s.t. \qquad W^*(A) = \operatorname{argmin}_W L(W, A, D_1) \tag{16}$$

This optimization problem is solved using one-step gradient descent and finite-difference approximation similar to the one described in Section 3.6. After convergence, the final architecture is determined by selecting the operation with the highest architecture weight for each search node. For computational and memory efficiency, DARTS initially searches for the architecture with only a few search layers. After the searching phase, there is a model retraining phase: a larger model (called *final model*) with more layers is composed by stacking the searched layer multiple times; then the final model is retrained on the combination of data splits $D_1$ and $D_2$. However, this approach introduces a discrepancy between the search phrase and retraining phrase, in terms of the number of layers in the models.

PDARTS Chen et al. (2019) addresses this limitation by progressively increasing the architecture's depth during the search phase. Initially, PDARTS trains the architecture with a few layers for several epochs, then refines the operation set to only include those with high architecture weights. This procedure is repeated, gradually increasing the number of layers to match the final model's layer number. However, this approach leads to increased computational costs.

## F  Further Analysis of Searched Architectures

In this section, we present the architecture searched by baseline method, DARTS, as shown in Figure 4, for further comparison with the architecture searched by our method, Ours-darts. We observe that both DARTS and Ours-darts result in cross attention modules, highlighting their effectiveness in the machine translation task. However, DARTS method leads to multiple feed-forward network (FFN) layers, which are absent in

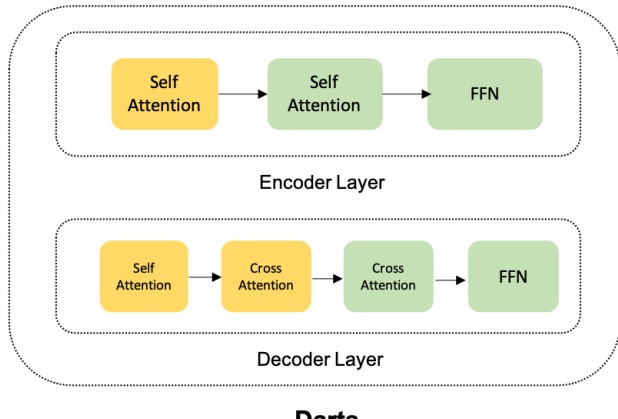

**Darts**

Figure 4: Architectures searched by darts on the WMT (En-De) dataset. The architecture consists of encoder layers and decoder layers. Multiple identical layers with searched architecture are stacked to construct the final model. Green boxes represent search nodes that are optimized during the search phase, while yellow boxes indicate predefined layers which are fixed during the search process.

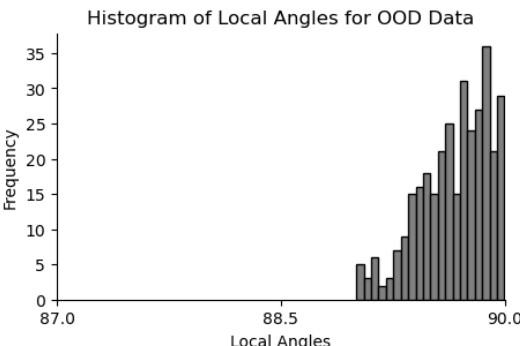

Figure 5: Distribution of local angles for generated approximated OOD samples.

architecture searched from Ours-darts. Instead, our method discovers GLU modules for the encoder layer, which can dynamically adjust their memory unit based on the input, potentially offering better adaptability to domain shifts compared to FFN layers.

## G   Analysis of Generated Samples in Stage II

In this section, we analyze some characters of the generated approximated OOD samples in the Stage II of our framework for improving the OOD performance of searched architecture. For each generated sample, we select 3 of the data samples in the real training dataset that are closest to this generated sample. These 3 samples determines a plane, and we compute the angle between the generated sample and this plane. The distribution of angles are presented in Figure 5, with an average degree of 89.66. Results show that the local angles between generated samples are close to 90 degree, which is almost vertical to the approximate plane of the training data. This provide a potential underlying reason for our method to achieve good out-of-domain generalization performance.

|            | En-Ig (Gnome)     | En-Ig (Ubuntu)    |
|------------|-------------------|-------------------|
| No-NAS     | $2.65 \pm 0.04$   | $2.15 \pm 0.05$   |
| Ours-darts | $\mathbf{2.97} \pm 0.01$ | $\mathbf{2.39} \pm 0.07$ |

Table 10: BLEU scores on two out-of-domain En-Ig datasets under an ablation setting: removing neural architecture search and only training a vanilla Transformer using synthetic out-of-domain data generated in Stage II.

## H    Impact of Neural Architecture Search

In this section, we conduct additional experiments to investigate the impact of neural architecture search in our method. We introduce an additional baseline by replacing the optimization of the trainable architecture in Stage III with optimizing model weights of a Transformer model instead (denoted as *No-NAS*). We present the results of this ablation study in Table 10. We observe that Ours-darts significantly outperforms the No-NAS baseline, highlighting the necessity of using neural architecture search. It also demonstrates that the out-of-domain samples generated in the second stage of our method do benefit the NAS process, instead of only being beneficial to model weights in general deep learning process.

