# OpenReview forum: "Transformer Architecture Search for Improving Out-of-Domain Generalization in Machine Translation"
_TMLR — Accepted by TMLR_

### Review · Reviewer_yu1j · 2024-08-23

**Summary Of Contributions:**

The paper proposes a multi-level optimization framework to search for transformer architectures, which is illustrated for machine translation. The main advance is the focus on the out of domain generalization, in which the authors generate artificial datasets and retrain and evaluate candidate models. The approach is evaluated on several datasets, showing superior OOD generalization performance, improving the state-of-the-art methods.

**Audience:**

Yes

**Broader Impact Concerns:**

Somewhat, please add a discussion section.

**Claims And Evidence:**

Yes

**Requested Changes:**

I believe this is a solid piece of work. I recommend accept as is and have minor requests:

1) Please address the weaknesses.

2) I am wondering about the angles that the ood samples have with the embedding manifold. Can you perhaps do one final study, in which you compute local angles between the manifold defined by the training data and the new added data? You can do so for example by picking (for each ood sample) the closest three real data points and computing an approximate plane, with respect to which you can compute ood data angles. I suspect ood data that have close to 90 degree angles will lead to better generalization, as they will be in orthogonal dimensions to existing data embeddings. Can you perhaps quantify this by using ood data for particular angles and showing us the effects? I think such an experiment can help you argue for the underlying reason for why this approach works and can lead to generalizable insights.

Edit: After reading other reviews, I sustain my belief that this work satisfies TMLR's criteria for acceptance.

**Strengths And Weaknesses:**

**Strengths**

- The experiments are convincing, methodological details are sufficient, and the results are therefore likely to be reproducible.
-  The paper is well motivated and addressed an important problem.
- The methodology is simple, yet elegant and significant. Thus, it can be presented to be accessible to a broader audience and can have further interesting applications.

**Weaknesses**

- I would have appreciated a proper discussion section, instead of sparse discussion points within results. This decreases the clarity of the writing.
- The related work section relates minimally back to current contributions. I think the distinctions should be made more clearly and the reader should be guided better. It is too generic in its current form.

---

> ### Author Response · Authors · 2024-10-19
> **Author Response**
>
> ## 1. Discussion Section
>
> Following the reviewer's suggestion, we have added a dedicated discussion section in the revised manuscript, incorporating key discussion points previously included in the results section. In this new section, we discuss the underlying reasons for the high in-domain performance of our method, provide a comparison of computational costs with baseline methods, and offer an analysis and comparison of the architectures discovered by our method against the baselines.
>
> ## 2. Related Works
>
> In accordance with the reviewer’s suggestions, we have revised the related work section. Content not directly relevant to our method, such as EA-based NAS and RL-based NAS, has been relocated to Appendix A. Moreover, in each subsection of the related work, we clearly articulate how existing methods relate to our approach and emphasize the key differences.
>
> ## 3. Analysis of Generated Samples
>
> We thank the reviewer for the insightful comments. In response, we conducted additional experiments to compute the angles between out-of-domain samples generated by our method and the embedding manifold of the training data. The results show that most angles are close to 90 degrees, with an average of 89.66 degrees. We have also created a barplot for the angle distribution in Figure 5 of the revised manuscript, in which the x-axis denotes the local angles and the y-axis shows the number of generated out-of-domain samples that fall in that local angle range. These findings provide further explanation for the superior performance of our method, as the generated samples are nearly orthogonal to the embedding manifold, enhancing generalization. A detailed discussion of this result has been added to Appendix G.

---

> > ### Comment · Reviewer_yu1j · 2024-10-19
> >
> > I would like to thank the authors for this revision. At a first blush, it seems the two weaknesses I raised are addressed, though I will take a one final detailed look later.
> >
> > I also thank you for running the experiment I requested. I believe the results are promising, but a null distribution is missing. In high dimensions, almost all vectors are approximately orthogonal to each other. Thus, I think a null distribution is necessary to argue that the nearly 90 degree angle is significant. One can generate such a null distribution by for example shuffling the entries of the perturbation. Then, you could run a rank-sum test on the difference between the null and true angle distributions and report the significance level.
> >
> > If the difference is significant, you can report it as a potential explanation for future work to chase. If not, you can report it as an interesting exploration, which was however not different than the null distribution. In either cases, I believe there is value in reporting this analysis. Thank you for your time!

---

> > > ### Author Response · Authors · 2024-10-23
> > > **Author Response 2**
> > >
> > > Thank you for your valuable feedback and positive suggestions. We appreciate your insightful comments regarding the null distribution for our angle analysis. We are conducting the experiment as you described and will update the manuscript with the results soon.

---

### Review · Reviewer_4pCy · 2024-09-06

**Summary Of Contributions:**

The authors propose a novel method which improves performance of existing methods in Neural Architecture Search (NAS), such as DARTS or PDARTS, by introducing a perturbation process into the multi-level optimization framework. The effectiveness of the proposed perturbation method in enhancing NAS performance is validated through comprehensive ablation studies. These studies demonstrate that optimizing the perturbation degree by maximizing the maximum mean discrepancy (MMD) is crucial for improving overall performance.

**Audience:**

Yes

**Broader Impact Concerns:**

The paper highlights the potential for performance improvement in NAS through appropriate perturbation methods.

**Claims And Evidence:**

Yes

**Requested Changes:**

The authors are encouraged to include recent references, particularly those exploring noise embeddings, such as Neftune (2023), to provide a broader context and strengthen the related work section.
Additionally, it is recommended to provide a comparative analysis of the architectures generated by DARTS and the proposed method to better illustrate the effects of the proposed perturbation technique.

Reference
- Jain, N., Chiang, P. Y., Wen, Y., Kirchenbauer, J., Chu, H. M., Somepalli, G., ... & Goldstein, T. (2023). Neftune: Noisy embeddings improve instruction finetuning. arXiv preprint arXiv:2310.05914.

**Strengths And Weaknesses:**

Strengths:

•	The paper presents a novel method of incorporating perturbation into embeddings, resulting in substantial performance improvements.

•	The authors demonstrate a well-optimized perturbation process using MMD, which further enhances overall performance.


Weaknesses:

•	The proposed method requires an additional search cost compared to baseline methods such as DARTS or PDARTS.

•	The current evidence does not fully elucidate how the perturbation process directly impacts the architecture obtained by NAS. It is possible that the observed performance improvements stem more from the perturbation process enhancing general deep learning training rather than specifically benefiting the NAS procedure itself. This distinction warrants further investigation to fully understand the mechanism of improvement.

•	While the paper provides valuable insights, there may be room to expand the literature review. Including a wider range of references, especially from recent work, could further enrich the paper's positioning within the field and potentially uncover additional connections to related research.

---

> ### Author Response · Authors · 2024-10-19
> **Author Response**
>
> ## 1. Search Cost
> Although our method introduces additional computational costs compared to baselines like DARTS and P-DARTS, the increase is modest. As presented in Table 8, the increase of time cost is on average less than 10%, which is acceptable in most cases as the marginal increase in computational expense is outweighed by the substantial performance gains observed across various OOD machine translation tasks.
>
> ## 2. Impact of NAS
>
> We perform an experiment to verify if the perturbation process directly impacts the architecture obtained by neural architecture search (NAS) or if perturbation is just improving the general deep learning optimization. To address this concern, we conducted an ablation study to isolate the impact of the perturbation on NAS from its effect on general model optimization. Specifically, we replaced the architecture optimization step in Stage III with direct model weight optimization of a Transformer, a setup we refer to as "No-NAS." In this configuration, the perturbation is applied directly to model training, bypassing the architecture search entirely. The results of this experiment, presented in Table 10 and Appendix H of the revised manuscript, show that our method significantly outperforms this baseline by 5.7%. These findings demonstrate that the perturbation process plays a critical role in enhancing the NAS, rather than merely improving general model training. This suggests that the observed improvements are indeed specific to the architecture search procedure.
>
> ## 3. Related Work
> We thank the reviewer for the valuable suggestions. In response, we have updated the related work section to provide a more comprehensive literature review. Specifically, we have added a new subsection (Section 2.3) to discuss recent advancements in noisy embeddings, and its connection to our proposed method.
>
> ## 4. Analysis of Searched Architectures
> In the revised manuscript, we provide the searched architecture of DARTS in Figure 4. We observe that both DARTS and Our method result in cross attention modules, highlighting their effectiveness in the machine translation task. However, DARTS method leads to multiple feed-forward network (FFN) layers, which are absent in architecture obtained by our methods. Instead, our method discovers GLU modules for the encoder layer, which can dynamically adjust their memory unit based on the input, potentially offering better adaptability to domain shifts compared to FFN layers. We include this additional discussion of searched architectures in Appendix F.

---

> ### Comment · Reviewer_4pCy · 2024-10-21
>
> I appreciate the effort the authors have put into preparing their rebuttal. However, I agree with the review GTSQ that a very low BLEU score may be critical against validating the significance. Therefore, I believe that more rigorous evidence is still necessary to demonstrate the significance of the proposed method.

---

> > ### Author Response · Authors · 2024-10-22
> > **Author Response 2**
> >
> > Thank you for your valuable feedback. Machine translation for low-resource languages is particularly challenging, especially in out-of-domain settings, which leads to the lower performance of both our methods and the baseline. We understand your concern regarding the low BLEU score and its potential impact on validating the significance of our approach. To address this, we conducted additional out-of-domain (OOD) evaluations on high-resource language pairs where larger training data is available, resulting in higher BLEU scores. Specifically, we trained our model on the WMT18 English-German (En-De) dataset and evaluated it on the WMT-Chat test set, which serves as OOD for the original training data. Our method achieved a BLEU score of 14.04 ± 0.14, compared to 13.28 ± 0.25 for the vanilla Transformer baseline, highlighting the effectiveness of our approach.
> >
> > These results are included in Appendix I, Table 11, where we further explore the OOD generalization performance. The higher BLEU scores in these high-resource settings reflect the benefits of abundant training data, contrasting with the challenges encountered in low-resource scenarios. Additionally, we are conducting further OOD experiments on other high-resource language pairs, which will be included in the final manuscript.
> >
> > We hope these additional evaluations address your concerns and demonstrate the robustness of our method across diverse settings.
> >
> > Thank you again for your insightful comments.

---

### Review · Reviewer_6KDA · 2024-10-03

**Summary Of Contributions:**

This work presents a novel approach for architecture search with transformers that is aimed at improving OOD generalization in machine translation. A 3-stage optimization procedure is proposed, involving first training a base model, then generating OOD data by learning a perturbation in embedding space, and finally updating the architecture so as to improve performance on the OOD data. The approach shows improvements for transfer generalization between different machine translation datasets.

**Audience:**

Yes

**Broader Impact Concerns:**

There are no broader impact concerns.

**Claims And Evidence:**

Yes

**Requested Changes:**

In what sense is the OOD test data out-of-distribution?

Can a control experiment be performed in which architecture search is replaced with standard learning methods (but which preserves the use of synthetic OOD data)?

Is the OOD test data in any way used during the architecture search procedure, or only used to evaluate the resulting architecture?

**Strengths And Weaknesses:**

I found this work to be very interesting, and the proposed optimization procedure is clever. It is especially interesting to target OOD generalization by learning to generate new OOD data. The general topic of OOD generalization in machine translation is also important, and the proposal seems to yield improved performance.

I do have a couple of questions and concerns regarding this work. First, the 'OOD' test data is drawn from different translation data for the same languages (e.g. training on CCMatrix and testing on Gnome and Ubuntu), but in what sense is this data out-of-distribution? Are there are any systematic differences between the training and test data, or is this setting labelled 'OOD' simply because they are from different datasets?

Second, my primary concern with the work is that, although the optimization procedure is clever, and seems to show improvements, it is not clear that these improvements are really specific to architecture search. I think an important control condition would be to use the same optimization procedure, but rather than using differentiable architecture search in the third step, simply update the base model. Are there really any benefits to optimizing the model through architecture search, rather than through standard learning methods? It is currently unclear whether the gains come from the use of architecture search vs. the method for generating synthetic OOD data.

Finally, I just want to clarify that the OOD test data is not in any way used during the optimization procedure, i.e. the only data used for architecture search is the synthetic OOD data, not the data that's ultimately used to evaluate OOD performance? (I believe this is the case, but want to confirm).

Minor note: the second paragraph of the intro uses the phrase 'DARTS-based' but this acronym has not yet been introduced.

---

> ### Author Response · Authors · 2024-10-19
> **Author Response**
>
> ## 1. Out-of-distribution Datasets
>
> We apologize for the confusion here. CCMatrix data and Gnome/Ubuntu data do share large domain discrepancies, not only because they are different datasets.
>
> Gnome contains technical and UI-related text specific to the Gnome environment, while Ubuntu contains system messages and interface text in Ubuntu OS, representing a software domain.The text in this domain includes specific terms ("permissions", "kernel", "GUI", "repository", "configuration file"), technical verbs ("execute," "compile," "deploy.") and abbreviations ("sudo," "bash," "root"). Sentences are typically task-oriented or informational, dealing with instructions, warnings, and descriptions of software functionality.
>
> In contrast, CCMatrix is derived from multilingual web pages which cover a wide range of topics, representing a general domain. For text in the general domain, it includes topics such as politics, sports, entertainment, and daily conversations. The word usage is diverse, including those related to politics ("democracy"), culture ("art gallery"), and emotions ("love"), which would be uncommon in software datasets.
>
> To sum up, the difference between CCMatrix and Gnome/Ubuntu datasets differs from multiple aspects including vocabulary and semantics, making them ideal datasets to investigate out-of-domain generalization performance of machine translation models. We have added discussion of the discrepancy between these two domains in Appendix D of revised manuscript.
>
> ## 2. Ablation Studies on Neural Architecture Search
>
> We thank the reviewer for the helpful suggestions. To address this concern, we conducted an ablation study to isolate the impact of the perturbation on NAS from its effect on general model optimization. We have conducted additional experiments to investigate the impact of neural architecture search by replacing the optimization step of architecture in the third level into the optimization of model weights of a Transformer (No-NAS). Experimental results show that our method outperforms the No-NAS baseline by 5.7%, indicating the necessity of neural architecture search. We put the experimental results in Appendix H in the revised manuscript.
>
> ## 3. Clarification and Minor Note
>
> We confirm that the OOD datasets are only used in the evaluation stage.
>
> We thank the reviewer for pointing out the minor note, and have revised that part of the sentence to ‘a differentiable neural architecture search method’ in the revised manuscript.

---

> > ### Comment · Reviewer_6KDA · 2024-10-26
> >
> > Thank you very much to the authors for the clarifications and additional ablation experiment. All of my concerns have been addressed.

---

### Review · Reviewer_GTSQ · 2024-10-07

**Summary Of Contributions:**

In this paper the authors present an approach for neural architecture search to improve machine translation on “out-of-domain” distribution data by finding architectures better suited to OOD  translation. The proposed solution builds on the DARTS framework based on differentiable search techniques. The key proposal is adding an additional intermediate stage in the multi-objective optimization that generates out of domain data, which is used for optimizing the last stage for architecture search.

**Audience:**

Yes

**Broader Impact Concerns:**

No broader impact concerns

**Claims And Evidence:**

No

**Requested Changes:**

* The paper needs a thorough overhaul of the evaluation setup as mentioned in weaknesses to be able to get interpretable results.
* Clarification of the conceptual weaknesses - how does perturbation result in out-of-domain data and samples of the same.
* An experiment of finetuning a base model with "OOD data" to see if finetuning with this data is sufficient to improve translation quality (in a revised experimental setup as suggested above).

**Strengths And Weaknesses:**

**Strengths**

The paper explores an interesting direction. Typically, for adapting to a new domain a pre-trained model is finetuned on domain specific data to modify the model parameters. This paper raises an interesting question of whether the architecture itself would be different for out-of-domain data?

**Weaknesses**

I will categorize the weaknesses broadly as conceptual weaknesses in the formulation of the solution and issues with the experimental setup.

_Conceptual weaknesses_

- The idea behind Stage 2 which generates synthetic data is to create out of domain data which will be used to train Stage 3. However, I am not sure that the proposed method used to create synthetic data results out of domain data.  All that is being done is that noise is added to the input embeddings and Stage 2 ensures that the model generates the same original translations on the target side (which means no change in semantics).  With no change in semantics, how does the generated data quality as out-of-domain data. What do the authors mean by out of domain data? My understanding would be there would be terms not encountered in the training data, or terms which have new meanings in the new domain, etc. How does the proposed approach ensure this?
- Adding noise with learned weights is probably some  smart regularization which might improve translation.
- There is not even a target domain that is specified - can you show some examples of the kind of sentences created with the perturbation (I suppose that is non-trivial).
- How do you justify the claim of “high-fidelity” OOD data – there is no evaluation of the quality of the synthetically generated data.
- In Stage 3, the training is done exclusively on supposedly “out of domain data” - no general domain data is used as far as I could understand. How can synthetic data alone lead to good performance on the translation task? Typically, domain adaptation is achieved by a  mix of general domain and domain-specific data. Furthermore, while the weights of the models are learnt on general domain corpora (stage 1), the architecture is learnt on "OOD" corpora (stage 3) - that does not sound right to me.
- Can you also mention how much synthetic data was created for Stage 3?
- Is there a degradation in general domain performance for the final model?

_Weaknesses in the experimental setup_

- Very low resource languages like Igbo and Hausa have been chosen for OOD evaluation. I am curious about the choice of these extremely low-resource languages to study OOD translation. It is difficult to get general domain performance at a decent level for these languages  - let alone performance in certain domains. I think this choice of OOD test data is not appropriate.
- The GNOME and Ubuntu datasets themselves are not great test sets - they are pretty noisy and not appropriate for evaluation.
- It would have been better to look at domain specific datasets in high resource languages The authors can take a look at the WMT shared tasks in many of the previous years for many domain specific datasets.
- **As the results show, the BLEU scores are very low (1-3) for OOD translation in low-resource languages. Those scores are basically meaningless and there is no translation happening – probably the source is just getting copied over.**
- Table #2: shows results of training on one language pair and testing on two different language pairs (English French and English Czech).  This can hardly be considered an out of domain experiment. These are cross lingual zero-shot translation experiments which are way more challenging than OOD translation.  I am skeptical of how such high BLEU scores for English to French/Czech translation were obtained by just training on English-German corpora since these are totally different languages. Something is not correct here.
- Table #4: It is not clear why the DARTS and PDARTS approach shows such degradation as compared to a vanilla transformer architecture – these approaches are supposed to work well for at least for in-domain scenarios and hence this is quite surprising. The proposed approach on the other hand seems to do better for in-domain scenarios though it was designed for out of domain scenarios. These observations of course hold only for Igbo and Hausa and not for German. Can you explain these? Testsets could themselves be an issue. CCMatrix is not a very clean corpus to use that as a test set, particularly for low-resource languages. A better testset would be the Flores-200 collection.

In the light of these major limitations in the experimental setup, it is difficult to establish the efficacy of the proposed approach.

---

> ### Author Response · Authors · 2024-10-19
> **Author Response (Part 1)**
>
> ## 1. What do the authors mean by out of domain data? How does the proposed approach ensure the generated synthetic data to be out of domain data?
>
> Thank you for your insightful comments. We appreciate the opportunity to clarify the definition of out-of-domain (OOD) data and explain how we generate synthetic OOD data in Stage 2. Our approach addresses the challenge where a model, trained on data from one domain, must perform well across different domains. In our method, we use the available training data to generate OOD data. We illustrate the concept of OOD data with the following example:
>
> - Daily speaking domain: "I gotta head out now, or I'll miss the bus."
> - Formal writing domain: "I must depart promptly to avoid missing the scheduled bus."
>
> Both sentences convey the same meaning but differ in tone and word choice, demonstrating a shift in distribution. Similar definitions of OOD data are used in computer vision, such as in [1], [2], and [3], where MNIST is used as the training domain, and datasets like SVHN serve as OOD data. Although both domains contain the same set of digits (0-9), they differ in distribution due to variations in handwriting styles and fonts.
>
> Based on this definition, we explain why our method can generate synthetic OOD data. We ensure semantic consistency between synthetic OOD samples and real data by optimizing the OOD samples to produce the same translations in the target language. At the same time, we maximize the distribution shift by increasing the Maximum Mean Discrepancy (MMD) loss between the sentence embeddings of the original and synthetic OOD samples. This creates a significant distributional difference while maintaining semantic similarity, aligning with the definition of OOD data provided earlier.
>
> We hope this explanation clarifies how our approach generates synthetic OOD data and addresses your concerns. Thank you again for your valuable feedback.
>
> [1] Volpi, R., Namkoong, H., Sener, O., Duchi, J., Murino, V., & Savarese, S. (2018). Generalizing to unseen domains via adversarial data augmentation. NeurIPS 2018.
>
> [2] Qiao, F., Zhao, L., & Peng, X. (2020). Learning to learn single domain generalization. CVPR 2020.
>
> [3] Qiao, F., & Peng, X. (2021). Uncertainty-guided model generalization to unseen domains. CVPR 2021.
>
> ## 2. Adding noise with learned weights is probably some smart regularization which might improve translation.
>
> We thank the reviewer for the insightful comment. Our current work focuses on improving the performance of neural architecture search methods on OOD data. We agree that adding noise to the learned weights of the model could be an effective regularization technique to enhance translation performance. We will explore this approach in future work to further boost the model's performance.
> We would also like to highlight that our method is different from adding noise to the training data as a regularization method. To verify this, we conducted an ablation study to evaluate the impact of removing Stage II, where we replaced the learned perturbations with random noise sampled from a Gaussian distribution, applied to the embeddings of real MT sentences to generate OOD samples. As presented in Table 4 of our updated manuscript, the model using learned perturbations outperforms the ablation model with random noise. This highlights the importance of learned perturbations using our method, which result in more effective OOD data generation compared to using arbitrary noise as regularization.
>
> ## 3. There is not even a target domain that is specified.
>
> As mentioned above, our goal is to simulate a worst-case generalization scenario, where the model must perform well across multiple unseen domains despite being trained on a single domain. Thus, we do not specifically choose a target domain.

---

> ### Author Response · Authors · 2024-10-19
> **Author Response (Part 2)**
>
> ## 4. How do you justify the claim of “high-fidelity” OOD data?
>
> We apologize for the confusion. Our intention was to indicate that the generated OOD sentence embeddings lead to improved performance on the test OOD data, which is why we referred to them as high-fidelity. However, we understand that this term may be misleading without a direct evaluation of the synthetic data quality. We have removed this term in the revised manuscript to only reflect the improvement in OOD performance without implying an unverified level of data quality.
>
> ## 5. How can synthetic data alone lead to good performance on the translation task? Why are the weights of the models learnt on general domain corpora (stage 1), but the architecture is learnt on "OOD" corpora (stage 3)?
>
> Thank you for your valuable feedback. Our framework consists of two phases: in the first phase, the architecture is searched to improve OOD performance, and in the second phase, the model is re-trained with the learned architecture on the entire training dataset.
>
> During the first phase for architecture search, we formulate the framework as a three-level optimization problem which consists of three stages. In the first stage of the architecture search, the model weights are trained on the original training domain data only. In the second stage, a perturbation is learned to maximize the domain difference between the OOD data and the original data while retaining the same translations (as mentioned earlier, we aim to improve OOD performance using only single-domain data, ensuring that the translation for the perturbed source sentence remains unchanged). In the third stage, the architecture is searched using the OOD data to further enhance OOD performance. If training domain data is included at this stage, both the architecture and model weights could overfit to the original domain, leading to reduced performance on OOD data. This issue is illustrated in the results from DARTS and PDARTS (Tables 1 and 4), which show diminished performance compared to the vanilla Transformer, as both architecture search and model parameter optimization were performed on the same training domain, resulting in overfitting.
>
> ## 6. Can you also mention how much synthetic data was created for Stage 3?
>
> Thank you for your question. We generate one OOD synthetic sample for each training instance in Stage 3. This ensures that the amount of synthetic data matches the size of the original training data. We hope this clarifies your query. Thank you again for your feedback.
>
> ## 7. The choice of low-resource languages for experiments on out-of-domain generalization.
>
> We thank the reviewer for the insightful comments on the experimental setup. We chose low-resource language pairs like Igbo and Hausa because they have limited parallel training data compared to high-resource languages, making out-of-domain (OOD) generalization a more prominent challenge in such settings. However, we acknowledge that the performance of models trained on these limited datasets is low.
>
> In response to the reviewer’s suggestion, we evaluated the OOD performance of our approach on high-resource language pairs. We trained our model on the WMT18 English-German (En-De) dataset and evaluated it on the WMT-Chat task, which is OOD for the training data. Our method achieved a BLEU score of 14.04 ± 0.14, compared to the vanilla Transformer baseline, which scored 13.28 ± 0.25. We have added an additional section in Appendix I to investigate the out-of-domain generalization performance of our method on high-resource languages, with results shown in Table 11.  While experiments on DARTS and PDARTS baselines are still ongoing, other results presented in the paper show that our method outperforms these baselines, and in some cases, these baselines even underperform the vanilla Transformer. We will update these results in the final draft.
>
> ## 8. GNOME and Ubuntu are noisy.
>
> We acknowledge that the GNOME and Ubuntu datasets are noisy. To address this concern, we are also conducting additional experiments on the Flores dataset, which provides cleaner data. The results on Flores dataset will be provided in the final version of our manuscript.

---

> ### Author Response · Authors · 2024-10-19
> **Author Response (Part 3)**
>
> ## 9. BLEU scores obtained for low-resource language pairs are very low.
>
> We acknowledge that the BLEU scores for low-resource language pairs are low, which can be attributed to the limited availability of parallel training data. To further illustrate the efficacy of our approach, we also conducted out-of-domain (OOD) evaluations on high-resource language pairs. Specifically, we trained our model on the WMT18 English-German (En-De) dataset and evaluated it on the WMT-Chat test set, which serves as OOD for the original training data. Our method achieved a BLEU score of 14.04 ± 0.14, compared to 13.28 ± 0.25 for the vanilla Transformer baseline, highlighting the effectiveness of our approach. The higher BLEU scores in these high-resource settings reflect the availability of significantly larger training data, which contrasts with the challenges faced in low-resource scenarios. We have added an additional section in Appendix I to investigate the out-of-domain generalization performance of our method on high-resource languages, with the results presented in Table 11.
>
> ## 10. High BLEU scores in Table 2 on translation for English-French and English-Czech.
>
> Thank you for your feedback. We would like to clarify that the experimental setups for Table 1 and Table 2 are different. In the setup for Table 1, we conducted an architecture search on a specific language pair (e.g., En-Ig) using the CCMatrix training domain. The learned architecture was then re-trained on the same training domain and evaluated on different domains (GNOME and Ubuntu) within the same language pair (En-Ig).
> In the setup of table 2, we first identified the optimal architecture by performing an architecture search on the English-German language pair. Once the optimal architecture was found, we retrained the model using this identified architecture on other language pairs, specifically English-French and English-Czech, using their respective corpora. This means that while the architecture was determined based on English-German data, the models for English-French and English-Czech were trained from scratch using the appropriate datasets for these language pairs. The goal of this experiment was to demonstrate that although the architecture was learned from one language pair, it generalized well and performed effectively on other language pairs. We have revised Section 4.4 and captions for Table 2 of the manuscript to avoid any confusion regarding the experimental settings and the BLEU scores.
>
> ## 11. Reasons for the DARTS and PDARTS approach show such degradation as compared to a vanilla transformer architecture in Table 4. A better testset would be the Flores-200 collection.
>
> Thank you for your feedback. The degradation in DARTS and PDARTS performance compared to the vanilla Transformer can be attributed to their sensitivity to noise, as both methods learn the architecture and weights on the same dataset. Since CCMatrix is noisy for low-resource languages like Igbo and Hausa, this likely caused overfitting. In contrast, WMT, being cleaner, led to better results for these methods. Our approach avoids this issue by performing architecture search using out-of-domain (OOD) data and training model weights on in-domain data, which helps prevent overfitting to noisy datasets. This likely explains why our method performs better on in-domain scenarios, despite being designed for OOD tasks. Regarding the test set, the experiments on Flores-200 dataset are running, and we will provide the results on this dataset as a part of our out-of-domain evaluations in the final manuscript.
>
> ## 12. An experiment of finetuning a base model with "OOD data" to see if finetuning with this data is sufficient to improve translation quality.
>
> We thank the reviewer for this suggestion. We perform additional experiments as an ablation study to verify if finetuning a base model with "OOD data" is sufficient to improve translation quality. Specifically, we replaced the architecture optimization step in Stage III with direct model weight optimization of a Transformer, a setup we refer to as "No-NAS." In this configuration, the perturbation is applied directly to model training, bypassing the architecture search entirely. The results of this experiment, presented in Table 10 of Appendix H of the revised manuscript. The results show that the No-NAS baseline outperforms the Transformer baseline without any OOD generalization method, while our method significantly outperforms the No-NAS baseline. These results support two key conclusions: (1) finetuning the base model with OOD data enhances its OOD performance, and (2) the perturbation process is crucial to improving the performance of the NAS-based method, beyond just improving general model training. This underscores the importance of the architecture search procedure in achieving optimal OOD performance.

---

> > ### Comment · Reviewer_GTSQ · 2024-11-06
> >
> > Thank you for your detailed response. Please find my comments on these responses below:
> >
> > 1. Thanks for your clarification. While this addresses some aspects of OOD generation - primarily synonymy, this is not very general in addressing polysemy, meaning shift, overloading of terms that might in OOD scenarios.
> > 2. The results in the Table 4 show very small differences, BLEU scores differences of 0.5 are unlikely to be statistically significant. So, even random noise seems to give similar performance. Hence, it is quite possible that the improvements are simply the result of regularization due to adding noise to the data.
> > 3. As mentioned in response to (2), it is difficult to say if there is a real OOD data generation happening or if this is just noise augmentation helping the model.
> > 4. ok, thanks for the clarification
> > 5. This 2 Phase training is not mentioned in the paper (including Figure 1), which only discusses the 3 stage architecture search. There is no mention of retraining on the original set after architecture search. As far as I understand, neither is that a standard approach in NAS. Please mention this in a revision as well as clarify/justify this departure from standard approach to NAS.
> > 6. Thanks for the clarification, please add the details to the paper too.
> > 7. Thanks, it is good to see the result on high resource languages. I think the main paper should report results on high resource languages only. Please add more OOD results on other domains/languages in the main paper. The low-resource results (Table 1) can be moved to the appendix. I don't even mind not reporting them. The OOD results should be reported on high-resource languages only - but we need to see results on more tasks and languages. The one result you have presented now is a good start.
> > 8. FLORES is a general-domain testset, not ideal for OOD evaluation. As I suggested please take a look at the WMT shared tasks over the years.
> > 9. Please see response to (7)
> > 10. Thanks for the clarification and revisions. That makes the results understandable. However, even in this case the baseline DARTS also works fine - the proposed approach only marginally improves within the limits of statistical significance mostly.
> > 11. I look forward to these results before drawing a conclusion.
> > 12. Results on these languages with such low BLEU scores are not useful. Request you to do these experiments on high resource languages.
> >
> > Overall, your responses and revisions have brought some clarity to my questions. However, 2 major issues remain which must be resolved to demonstrate effectiveness of this approach:
> > - Please reports experiments on high resource languages on multiple domains for comprehensive evaluation. The results on low-resource languages with near-zero BLEU scores are not useful.
> > - It is not clear that the improvements due to the MMD data generation stage are statistically significant from adding noise during training.

---

> > > ### Author Response · Authors · 2024-11-18
> > > **Author Response**
> > >
> > > We thank the reviewer for the constructive feedback. In response to the concerns, we have conducted additional experiments and provided the results below.
> > >
> > > **1. Additional OOD Results on High-Resource Languages**
> > >
> > > We have evaluated our method and the baseline on the out-of-domain (OOD) English-German translation task in the biomedical domain using the Himl Test Sets (WMT-24 Shared Task: Biomedical Translation Task). The results, averaged over 3 runs, are presented below:
> > >
> > > | Method     | BLEU     |
> > > |--------------|--------------|
> > > | Transformer  | 54.08 ± 0.11  |
> > > | **Ours-darts**  | **58.71 ± 0.46**  |
> > >
> > > Furthermore, we evaluated our method on the OOD English-French translation task in the chat domain using the WMT-Chat dataset. The results, averaged over 3 runs, are presented below:
> > >
> > > | Method     | BLEU     |
> > > |--------------|--------------|
> > > | Transformer  | 26.90 ± 0.58  |
> > > | **Ours-darts**  | **28.06 ± 0.20**  |
> > >
> > > These results demonstrate that our proposed method consistently outperforms baseline methods on OOD tasks across various domains and language pairs, highlighting its robustness and effectiveness in the setting of high-resource languages. We are currently conducting additional experiments across more domains and languages to provide a more comprehensive evaluation of our approach.
> > >
> > > **2. Effectiveness of the MMD Data Generation Stage**
> > >
> > > To verify the statistical significance of our method compared to that with random noise, we conducted additional runs (3 runs) using different random seeds for both our method (Ours-darts) and the baseline method under this ablation setting (No-Stage-II-darts). The results, presented in terms of BLEU score, are shown below:
> > >
> > > | Method     | En-Fr     | En-Cs     |
> > > |--------------|--------------|--------------|
> > > | No-Stage-II-darts  | 38.57 ± 0.07  | 25.01 ± 0.07  |
> > > | **Ours-darts**  | **38.88 ± 0.10**  | **25.31 ± 0.05**  |
> > >
> > > We observe that, although the absolute difference between our method and the baseline is small, our method consistently outperforms the baseline across multiple runs, with a standard deviation of less than 0.1. These results highlight the effectiveness of the MMD data generation stage with statistical significance.
> > >
> > > As suggested by the reviewer, we will include all the results mentioned above in the main text of our paper. Thank you again for engaging with us and your valuable feedback. We would be delighted to answer any further questions you might have.

---

### Decision · Action_Editor_XTYf · 2024-11-17

**Recommendation:** Accept with minor revision

**Comment:**

Shared above.

**Audience:**

It is appropriate for TMLR audience.

**Claims And Evidence:**

The paper presents an optimization approach for architecture search in machine translation (MT) for out-of-domain (OOD) scenarios (Figure 1). As part of this approach, an OOD MT dataset is generated to enhance generalization. This dataset is created through noise perturbation. The multi-objective framework (Section 3.4, Algorithm 1) seems to be a logical approach, supported by experiments to substantiate its benefits.

While all the reviewers appreciate the simplicity and logic of the approach, they have raised some critical comments regarding the experimental setup and the validity of the claimed benefits. Given that the experiments are the primary selling points, addressing these concerns is essential. Specifically, reviewers have pointed out:

1. The OOD generation step might not actually be producing genuine OOD data. While the OOD definition has been partially addressed in the author rebuttal, it would be prudent to clarify this further and downplay the OOD generation as a key contribution. This concern is shared by at least three reviewers.

2. The experimental setup's weakness, especially regarding low-resource languages. While the approach may perform well in other scenarios, the primary benefit of MT is typically seen in low-resource settings. Therefore, considering experiments that solidify these contributions would be beneficial.

Additionally, I have my own comment on the optimization aspect. While the idea of bilevel optimization is intriguing, is there any guarantee or supporting experiments to ensure that the overall search does not yield trivial solutions? For instance, does Algorithm 1 avoid converging to suboptimal points? Any comments on those would be great.

Based on these comments, I believe the paper requires revisions. I would expect the revised version to sufficiently address the above concerns.

**Resubmission Of Major Revision:**

The authors may consider submitting a major revision at a later time.